# Beneficial Effects of Dietary Fiber in Young Barley Leaf on Gut Microbiota and Immunity in Mice

**DOI:** 10.3390/molecules29081897

**Published:** 2024-04-22

**Authors:** Seita Chudan, Takuto Kurakawa, Miyu Nishikawa, Yoshinori Nagai, Yoshiaki Tabuchi, Shinichi Ikushiro, Yukihiro Furusawa

**Affiliations:** 1Department of Biotechnology, Faculty of Engineering, Toyama Prefectural University, Toyama 939-0398, Japanikushiro@pu-toyama.ac.jp (S.I.); 2Department of Pharmaceutical Engineering, Faculty of Engineering, Toyama Prefectural University, Toyama 939-0398, Japan; 3Division of Molecular Genetics Research, Life Science Research Center, University of Toyama, Toyama 930-0194, Japan

**Keywords:** young barley leaf, dietary fiber, dendritic cell, Dectin-1, gut bacteria, short-chain fatty acid, retinoic acid, IgA

## Abstract

The health benefits of young barley leaves, rich in dietary fiber, have been studied for several decades; however, their beneficial effects on the intestinal microenvironment remain to be elucidated. To investigate the effects of young barley leaf-derived dietary fiber (YB) on the gut microbiota and immunity, mice were fed an AIN-93G diet containing cellulose or YB and subjected to subsequent analysis. The population of MHC-II-positive conventional dendritic cells (cDCs) and CD86 expression in the cDCs of Peyer’s patches were elevated in the YB-fed mice. MHC-II and CD86 expression was also elevated in the bone marrow-derived DCs treated with YB. 16S-based metagenomic analysis revealed that the gut microbiota composition was markedly altered by YB feeding. Among the gut microbiota, Lachnospiraceae, mainly comprising butyrate-producing *NK4A136* spp., were overrepresented in the YB-fed mice. In fact, fecal butyrate concentration was also augmented in the YB-fed mice, which coincided with increased retinaldehyde dehydrogenase (RALDH) activity in the CD103^+^ cDCs of the mesenteric lymph nodes. Consistent with elevated RALDH activity, the population of colonic IgA^+^ plasma cells was higher in the YB-fed mice than in the parental control mice. In conclusion, YB has beneficial effects on the gut microbiota and intestinal immune system.

## 1. Introduction

The immune system is a defense mechanism against infectious pathogens (e.g., viruses, bacteria, and parasites). Mucosal tissue in the respiratory and digestive organs is a route of infection for pathogens because the mucosa is the interface between the host and foreign substances. Mucosal epithelial cells and mucin form a physical barrier, and antibacterial peptides secreted by Paneth cells act as chemical barriers by killing pathogens to prevent their invasion [1]. If the pathogens penetrate these barriers, the mucosal immune cells generate defense responses against foreign pathogens.

The immune system can be divided into the innate and acquired immune systems. The innate immune system, which includes neutrophils, macrophages, and natural killer (NK) cells, enforces immediate but nonspecific responses, whereas the acquired immune system elicits a specific response to antigens [2]. As the acquired immune system in the mucosa, dendritic cells (DCs) recognize, process, and present antigens to the B and T cells against pathogen infection, leading to the production of immunoglobulin A (IgA) from B cell-differentiated plasma cells, as well as the activation of killer and helper T cells, which play a pivotal role in defense against invading pathogens [3].

DCs are functionally classified as classical DCs (cDCs) and plasmacytoid DCs (pDCs) [4]. cDCs induce type 1 helper T cell differentiation and NK cell activation through IL-12 production. Additionally, cDCs promote the immunoglobulin A (IgA) class switch in B cells by producing retinoic acid [5]. cDCs activation prevents the progression of infectious diseases by educating the acquired immune system. Thus, functional materials that activate cDCs have attracted attention among researchers, healthcare workers, and consumers.

To identify functional foods that are beneficial to the immune system, we have previously investigated the effects of grain-derived materials rich in dietary fiber on the gut microenvironment [6,7]. This study aimed to investigate whether young barley leaf-derived dietary fiber (YB) is a candidate functional material from the perspective of intestinal immunity. Young barley leaf powder, which is rich in dietary fiber, is produced from the stems and leaves of barley plants after washing, drying, and crushing. Recently, young barley leaf powder suspended in water, called Aojiru, has been consumed worldwide as a beverage to promote human health. This powder, and in particular YB, improves bowel movements and inhibits hyperlipidemia and diabetes [8,9]. 

Dietary fiber is a mixture of polysaccharides, including ten or more monosaccharides, and is not hydrolyzed by human gastrointestinal enzymes [10]. The indigestible polysaccharides include cellulose, hemicellulose, and lignin. Hemicellulose, a generic term for polymers composed of diverse sugars (e.g., glucose, xylose, galactose, mannose, rhamnose, and arabinose) in plant cell walls, is recognized by the membrane receptors (e.g., pattern recognition receptors) expressed in intestinal epithelial and myeloid cells (e.g., macrophages and DCs), leading to the modification of cellular functions [11]. Among the receptors for indigestible polysaccharides, Dectin-1, a C-type lectin, has been well studied as an immunomodulatory receptor [12]. Dectin-1 binds to 1,3-1,4-β-glucan and arabinoxylan [13]; the former comprises a hemicellulose component of the dietary fiber in oat and barley grains [14]. The latter was observed in barley leaf and wheat bran, although the number of arabinose side chains in barley arabinoxylan is greater than that in wheat bran [15]. Several studies have demonstrated that Dectin-1 agonists promote the functional maturation of myeloid cells, as exemplified by elevated cytokine production (e.g., IL-6 and IL-12) and the expression of major histocompatibility complex II (MHC-II) and co-stimulatory molecules (e.g., CD80 and CD86) [16] (also reviewed in [12]).

Dietary fiber is largely consumed by commensal bacteria abundantly present in the large intestine, resulting in a change in the gut bacterial composition [17]. Among the constituents of dietary fiber, hemicellulose is more readily fermentable than other low-fermentable fiber. During fermentation, certain gut bacteria produce metabolites such as short-chain fatty acids (SCFAs), which modulate the immune functions of both lymphocytes (e.g., T and B cells) and myeloid cells (e.g., neutrophils, macrophages, and DCs) [18,19]. Accumulating evidence has demonstrated that butyrate maintains the gut mucosal immune system, as exemplified by retinoic acid production through the enhancement of retinaldehyde dehydrogenase (RALDH) activity in cDCs [20], which leads to IgA class switching in B cells [21]. 

Considering the aforementioned considerations, the intake of YB is anticipated to have beneficial effects on the gut microbiota and mucosal immune system. In support of this hypothesis, young barley leaf, rich in fiber, has recently been reported to increase *Bifidobacterium* spp. and butyrogenic bacteria in human colonic microbiota cultures [22]. However, the beneficial effects of YB on intestinal immunity and the association with changes in the gut microbiota remain to be elucidated. 

In the present study, we evaluated the effects of YB on intestinal immunity and the gut microbiota in mice fed chow containing YB. We demonstrated that YB modified cDC function in both the Peyer’s patch (PP) and mesenteric lymph nodes and augmented IgA^+^ plasma cells in the colonic lamina propria, the latter of which may be a result of increased butyrate-producing bacteria and luminal butyrate levels.

## 2. Results

### 2.1. YB Modifies cDC Function in the PP

Orally administered indigestible polysaccharides pass through the small intestine before reaching the large intestine. During this process, a small portion of indigestible polysaccharides is believed to enter the small intestinal tissue through the PP.

The PP is a lymphoid follicle composed of abundant leukocytes under the epithelium on which microfold cells (M cells) act as transporters of microbes and particles for immunosurveillance [23]. To determine the impact of YB on the expression of functional markers in PP cDCs, we isolated PP cells from mice fed AIN-93G containing cellulose (control: CT) or YB. In the dissection and cell isolation processes, we noticed that the PP size and cell numbers were augmented by YB feeding (Figure 1A,B). The percentage of CD11c^+^/MHC-II^+^ cells showing mature cDCs was also increased by YB feeding (Figure 1C,D). Among the mature PP cDCs, the expression of CD86, a co-stimulatory molecule on antigen-presenting cells, was elevated in the YB-fed mice (Figure 1E,F).

### 2.2. YB Activates Bone Marrow-Derived DCs (BMDCs)

To examine whether YB has the potential to directly modify DC function, we quantified functional markers of BMDCs treated with YB for 1 d. In line with expectations, we detected increases in the percentages of CD80^+^/86^+^ (both of which are co-stimulatory molecules) cells and the expression of MHC-II in CD11c^+^ BMDCs in response to YB treatment (Figure 2A,B). Moreover, cells treated with YB and young barley leaf powder were found to show comparable rates of induction (Appendix A), indicating that the dietary fiber of young barley leaves influences the functional maturation of BMDCs. A previous study demonstrated that YB comprises CE and hemicellulose (including β-glucan and particularly arabinoxylan) [9]. Considering that approximately 50% of YB comprises CE (51.8% of total fiber, as shown in [9]), we subsequently examined the effects of CE on the expression of CD80/86 and MHC-II in BMDCs, in which the amount of cellulose in the culture medium was adjusted to be slightly lower or higher than that of YB. In contrast to YB, we found that none of the assessed concentrations of CE had any appreciable effects on the populations of CD80/86 BMDCs or the expression of MHC II (Figure 2A,B).

Consistent with the surface marker expression, we observed that YB promoted the secretion of IL-12 and IL-6 from BMDCs (Figure 2C), and that the IL-12 production induced by YB was abrogated by pretreatment with whole glucan particle (WGP), an antagonist of Dectin-1 (Appendix A). These results indicate that the indigestible polysaccharide in YB, which acts as a Dectin-1 agonist and is probably the hemicellulose fraction of YB, contributes to DC activation. Consistent with this supposition, we demonstrated that low molecular weight β-glucan, an agonist of Dectin-1, had no additive or synergistic effects on the expression of CD80/86 or MHC-II in BMDCs treated with YB (Appendix A). 

To confirm the function of BMDCs treated with YB, these cells were co-cultured with naïve CD4^+^ T cells in the presence of anti-CD3e antibody for T cell receptor stimulation. In line with expectations, we determined that YB promoted the differentiation of IFN-γ-producing Th1 cells in a dose-dependent manner (Figure 2D,E). 

### 2.3. YB Modifies the Gut Microbiota

Dietary fiber reaches the colon and is consumed by gut commensal bacteria, leading to changes in the diversity and composition of the gut microbiota. To analyze the effect of YB on the gut microbiota, we collected feces from mice fed the CT or YB for 4 weeks, followed by 16S rRNA amplicon sequencing. In this experiment, we first analyzed *α*- and *β*-diversity to compare the number of observed species and the microbial composition in CT- or YB-fed mice. The *α*-diversity analysis showed a slight, but not significant, change in the number (observed species and Chao1) and evenness (Shannon and evenness) of microbial species in the mice fed YB (Figure 3A and Appendix A). On the contrary, the *β*-diversity analysis revealed that the bacterial composition patterns were markedly distinct between the CT and YB groups (Figure 3B), indicating that the bacterial composition, rather than the number and evenness of species, was altered by YB feeding. Next, to identify the altered gut microbiota in the YB-fed mice, we performed taxonomic classification of the 16S rRNA-targeted amplicons. In the taxonomic analysis at the phylum level, Firmicutes and Bacteroidota (synonym Bacteroidetes) were the dominant communities in both groups. Among these phyla, YB feeding did not significantly affect the relative abundances of Firmicutes and Bacteroidetes (Figure 3C and Appendix A). Proteobacteria, a major phylum of Gram-negative bacteria comprising several pathogenic bacteria [24], were significantly less abundant in the YB-fed mice (Appendix A).

In the taxonomic analysis at the family level, Lachnospiraceae and Ruminococcaceae, which are composed of butyrate-producing bacteria [25], were overrepresented in the YB-fed mice (Figure 4A). In the taxonomic analysis at the genus and species levels, the genera belonging to Lachnospiraceae were more predominant than the other genera (Figure 4B). Among the Lachnospiraceae, the *Lachnospiraceae NK4A136 group* spp., *Eubacterium xylanophilum group* spp., and *Lachnospiraceae UCG-006* were overrepresented in YB-fed mice. As for the bacteria belonging to Ruminococcaceae, *Eubacterium siraeum group* spp. and *Incertae_Sedis Closrtidium* sp. were abundant in the YB-fed mice, although the relative abundances of the two species were subordinate (Figure 4B). In addition, the relative abundance of Prevotellaceae was found to be elevated in response to feeding with YB. Among the Prevotellaceae [26], *Prevotellaceae*_ *UCG-001* spp., previously reported to be associated with propionate production [27], were found to be over-represented in the YB-fed mice (Figure 4A,B).

### 2.4. YB Augments Fecal SCFAs

Dietary fiber is fermented by certain gut commensal bacteria in the colonic lumen, leading to a change in the abundance of fermentable metabolites such as SCFAs [18]. Given that YB increases the relative abundance of SCFA-producing bacteria, we quantified the SCFA concentrations in the feces collected from the CT- and YB-fed mice. Consistent with the elevated levels of SCFA-producing bacteria, fecal acetate, propionate, and butyrate levels were significantly elevated in the YB-fed mice (Figure 5). 

### 2.5. YB Promotes RALDH Activity in Mesenteric Lymph Node (MLN) cDCs

Butyrate has been reported to modify gut immunity by altering cDC function [20,21,28]. In this study, we assessed the effects of YB on the surface marker expression and RALDH activity in MLN cDCs. CD11c^+^/MHC-II^+^ cDCs were classified into at least three groups: CD103 single-positive (SP) cDC1, CD11b SP, and CD103/CD11b double-positive (DP) cDC2. Previous studies demonstrated that CD103 SP cDC1 and CD11b SP cDC2 were more predominant than DP cDC2 [29]. In addition, RALDH activity in the CD103^+^ cDCs has been known to be higher than that in CD11b SP cDC2 [30]. Consistent with a previous report, the percentage of SP cDCs was higher than that of DP cDCs, and RALDH activity in the CD103^+^ cDCs was dominant (Figure 6A–D). The composition of SP and DP cDCs was comparable between the CT- and YB-fed mice (Figure 6A,B); however, YB feeding promoted RALDH activity in CD103 SP cDC1 (Figure 6C,D).

### 2.6. YB Augments IgA^+^ Plasma Cells in Colonic Lamina Propria

Butyrate was reported to increase colonic IgA^+^ cells through the elevation of RALDH activity in cDCs [21], which metabolizes retinol to all-trans retinoic acid, thus enhancing the T cell-independent IgA response [5]. To evaluate whether elevated RALDH activity in CD103^+^ cDCs of YB-fed mice contributes to the functional modification of intestinal immunity, we quantified colonic IgA^+^ plasma cells in YB-fed mice. Consistent with the results of a previous report, the majority of IgA^+^ cells in the colonic CD3-negative population were B220-negative plasma cells (Figure 7A,B). As expected, the percentage of IgA^+^ cells in the B220-negative (plasma cell population) but not-positive fraction (B cell population) was increased by YB feeding (Figure 7A,B), indicating that the dietary fiber of young barley leaves modifies gut immunological function through alterations in the gut microbiota and its metabolites.

## 3. Discussion

In the present study, we evaluated the immunomodulatory effect of YB on the intestinal immune system from two aspects: the direct effect on the PP in the small intestine and the indirect effect through changes in the colonic microenvironment. Although microbiota is present in the lumen of the small intestine, the number of bacteria in the small intestine is much lower than that in the large intestine [31,32]. In addition, indigestible polysaccharides are unlikely to be fermented under aerobic conditions in the lumen of the small intestine. Therefore, the effect of YB on PP development appears to depend on its physical properties rather than on fermentation. A previous study reported that cornhusk arabinoxylan enhances PP development in mice [33]. In addition, birds fed a diet rich in dietary fiber were reported to develop a PP in comparison with those fed a low-fiber diet [34]. Furthermore, lentinan, an indigestible polysaccharide isolated from *Lentinus edodes* (mushroom) increases the PP size and promotes the proliferation of PP leukocytes [35]. These results suggest that YB is likely implicated in the increased PP size and number of PP cells, although the mechanism underlying the development of the PP by dietary fiber has not been revealed.

As shown in Figure 2, YB was responsible for DC activation. It has been suggested that indigestible polysaccharides, including β-glucan particles, can be transported into the PP through M cells and activate immune cells [36,37]. Among the indigestible polysaccharides in young barley leaf, the hemicellulose fraction is a promising candidate for cDCs activation since both β-glucan and arabinoxylan, typical components of hemicellulose, are known to activate human DCs in vitro by binding to Dectin-1 [13]. In support of this, a Dectin-1 antagonist counteracted IL-12 production in BMDCs with YB treatment (Appendix A). These results indicate that dietary fiber in young barley leaves, particularly the hemicellulose fraction, activates cDCs through Dectin-1, at least in part.

However, the association between YB and PP cDCs activation in vivo remains unknown. It is possible that PP cDCs activation is partly accompanied by the enhanced maturation of the PP, as well as direct stimulation by dietary fiber in young barley leaves. Therefore, the effect of YB on the small intestinal epithelial cells or PP formation should be evaluated using an organoid culture system [38]. Further investigations are required to reveal the effects of YB on lymphoid tissue formation in the small intestine.

In addition to the functional modification of cDCs, our further interest is the effect of YB on pDCs and NK cells. pDCs are a subset of DCs that secrete type I interferons in response to viral infections [4]. Although the expression level of Dectin-1 in pDCs are lower than that in cDCs [39], it is possible that YB activates both pDC and cDC via Dectin-1. NK cells, a population of cytotoxic lymphocytes, rapidly kill virus- or pathogen-infected cells missing MHC-I [40]. Although the expression of Dectin-1 in NK cells is minimal, IL-12 produced by cDCs is known to augment NK cell activity. Thus, YB might indirectly activate NK cell function by enhancing IL-12 secretion from cDCs. The effects of YB on pDC and NK cell function in vitro and in vivo will be investigated in future studies.

Hemicellulose modifies the gut microbiota and alters fermented metabolites in the large intestine. In the present study, YB feeding augmented the abundance of some genera belonging to Lachnospiraceae (Figure 4B); among these, the *Lacnospiraceae NK4A136 group* spp. are butyrogenic bacteria [41]. Our previous study demonstrated that these genera were overrepresented in mice fed wheat-derived arabinoxylan [6]. Considering that a greater part of the hemicellulose in young barley leaves also comprises arabinoxylan [42], *Lachnospiraceae NK4A136 group* spp. may become predominant by consuming the arabinoxylan in YB, despite the presence of a slight structural difference between barley and wheat arabinoxylans [15]. Among the other Lachnospiraceae augmented in the YB-fed mice, *E. xylanophilum* is a potent butyrate-producing bacterium. Previous studies have demonstrated that polysaccharides from wheat bran stimulate the production of butyrate in the human microbiota by promoting the growth of *E. xylanophilum* [43,44]. The overrepresentation of these genera may be responsible for the elevated luminal butyrate levels in YB-fed mice (Figure 5). Although Ruminococcaceae comprises butyrate-producing bacteria, it is unknown that *Incertae sedis* spp. produce butyrate. A previous report denied butyrate production by *Clostridium incertae sedis* (not Ruminococcaceae) [45]. *Eubacterium siraeum group* spp. were also overrepresented in the YB-fed mice; however, there have been no reports investigating butyrate production from this genus. Microbial cultures in the presence of indigestible polysaccharides should be performed to determine their butyrate-producing abilities. 

Butyrate has been reported to modify immune cell function through histone deacetylase inhibition and G-protein-coupled receptor activation in dendritic cells [20,21,28]. In the present study, YB promoted RALDH activity in CD103^+^ cDC1 cells without affecting the population of cDC subsets (Figure 6). Although it is unknown why YB did not affect RALDH activity in CD103^+^ cDC2 (DP), retinoic acid production in the total CD103^+^ cDCs population was augmented by YB feeding, as the DP cDCs population was minor in comparison to the SP cDCs (Figure 6A,B). In support of the elevated RALDH activity, YB feeding increased the population of IgA^+^ plasma cells (Figure 7), which is consistent with a previous report demonstrating that butyrate upregulated the production of all-trans retinoic acid from cDCs and thus led to an IgA class switch in B cells [21]. In future studies, the functional significance of increased IgA^+^ cells in YB-fed mice should be confirmed using an infection model. 

Our further interest is the effect of YB on the induction of regulatory T cells. Regulatory T cells, a T cell subset implicated in immune tolerance, maintain gut immune homeostasis by suppressing excessive immune responses against gut bacteria [46]. We previously demonstrated that both wheat-derived arabinoxylan and oat-derived soluble fiber rich in β-glucan augments the population of colonic peripherally induced regulatory T cells [6,7], leading to the suppression of T cell-dependent colitis. Both butyrate and retinoic acid contribute to the differentiation of naïve T cells into regulatory T cells in the colon [20,47]; YB might be a promising material even for the prevention of colitis onset through its ability to cause changes in the gut microenvironment including regulatory T cells induction. We plan to test this hypothesis in our future studies.

In conclusion, we demonstrated that YB has beneficial effects on intestinal immunity in mice, part of which may be the direct activation of cDCs and the indirect effects through changes in the gut microbiota and their metabolites. Although this study has limitations in terms of the animal experiment, we propose that young barley leaves may be a promising functional material for the modulation of intestinal immunity. 

## 4. Materials and Methods

### 4.1. Animals

C57BL/6J male mice (5–7 weeks old) were purchased from Japan SLC (Hamamatsu, Japan) and maintained under standard temperature, humidity, and light conditions (20–25 °C, 50 ± 10%, 12 h light/dark cycle, respectively). The mice were fed AIN-93G (Oriental Yeast, Tokyo, Japan) for an acclimation period of 1 week. In the modified diet, the CE in AIN-93G was replaced with YB (supplied by Toyo Shinyaku, Saga, Japan). The fiber content of the modified AIN-93G diet containing YB (comprising 64.5% dietary fiber) was adjusted to be equivalent to that of the original AIN-93G diet. The mice were fed YB for 4 weeks, followed by subsequent analysis.

### 4.2. Preparation of Cells from the PP, MLN, and Colonic Lamina Propria

The PPs were dissected, minced, and dissociated using 5 mL of RPMI1640 (Wako, Osaka, Japan) containing 0.5 mg/mL collagenase IV (Roche, Basel, Switzerland) and 0.5 mg/mL DNase I (Roche), by shaking (250 rpm) at 37 °C for 15 min. MLNs were incubated with a medium containing 0.5 mg/mL DNase for 15 min and then ground on a 100 mm mesh using a syringe plunger [47,48]. Colonic lamina propria cells were isolated as previously described [6,7,47,48,49]. Briefly, the colonic tissues were treated with 30 mL of Hank’s balanced salt solution containing 20 mM EDTA and 1 mM DTT at 37 °C for 20–30 min to remove the epithelial cells. The tissues were then minced and dissociated with 30 mL of the dissociation medium (0.5 mg/mL Collagenase IV and DNase I) at 37 °C, for 20–30 min, to obtain a single-cell suspension. The suspension was filtered, washed with RPMI 1640, and separated using a 40/80% Percoll gradient.

### 4.3. Generation of BMDCs and Treatment

Bone marrow cells in the femur were pushed out into PBS using a 1 mL syringe with a 26G needle. The collected cells were cultured in an RPMI-1640 medium containing 10% fetal bovine serum (FBS), 1% penicillin/streptomycin (Wako), 55 μM mercaptoethanol (Nacalai Tesque, Osaka, Japan), 10 mM HEPES (pH 7.2; Nacalai), and granulocyte–macrophage colony-stimulating factor (GM-CSF, 20 ng/mL; R&D systems, Minneapolis, MN, USA) for 6 days. An equal volume of complete medium containing 40 ng/mL GM-CSF was added to each well on day 3. After 6 days of culture, non-adherent cells were collected as the DC-enriched cell population. BMDCs were treated with LPS (InvivoGen, San Diego, CA, USA) or YB, followed by subsequent analysis. The YB was prepared from young barley leaf powder (Toyo Shinyaku) as previously described [50]. Cellulose and hemicellulose in the total fiber were 51.8% and 36.9% in the YB, respectively [9]. As a Dectin-1 antagonist, whole glucan particle (WGP) (InvivoGen) was used at the indicated concentrations.

### 4.4. Flow Cytometry

Cells were stained with monoclonal antibodies (mAbs) against CD3, CD4, CD11b, CD11c, CD45R/B220, CD80, CD86, CD103, IgA, or MHC-II (BioLegend, San Diego, CA, USA), followed by incubation with an anti-CD16/32 antibody (BioLegend). The antibodies were conjugated to FITC, PE, PerCPCy5.5, PE-Cy7, APC, APC-Cy7, Brilliant Violet 421, Brilliant Violet 605, or Brilliant Violet 711. Dead cells were distinguished using SYTOXA Advanced (Thermo Fisher Scientific, Waltham, MA, USA). RALDH-expressing cells were stained using an ALDEFLUOR kit (Stem Cell Technologies, Vancouver, BC, Canada) according to the manufacturer’s instructions. For intracellular cytokine staining, the cells were cultured for 4 h in a complete medium supplemented with 50 ng/mL PMA, 500 ng/mL ionomycin, and 5 μg/mL brefeldin A (Sigma, Burlington, MA, USA). The cells were harvested and stained with monoclonal antibodies against CD3 and CD4, followed by fixation in a fixation/permeabilization buffer (BioLegend). The fixed cells were subjected to intracellular staining with monoclonal antibodies against IFNγ- and IL-17A (BioLegend). Dead cells were distinguished using GhostDyeRed780 followed by fixation (Cytek Bioscience, Fremont, CA, USA). Stained samples were analyzed using a NovoCyte Flow Cytometer (ACEA Biosciences, San Diego, CA, USA) or Attune NxT (Thermo Fisher Scientific) and the FlowJo software version 10 (TOMY Digital Biology, Tokyo, Japan).

### 4.5. Cytokine Measurement

To quantify the cytokine concentration, the culture medium was collected and centrifuged, and the supernatant was used for subsequent analyses. A Mouse IL-12 p40 ELISA kit (Abcam, ab236717) or Cytometric Beads Array (CBA) kit (BD Bioscience, Tokyo, Japan) for IL-12 p40 and IL-6 were used according to the manufacturer’s protocol. 

### 4.6. Co-Culture of BMDCs with Naïve CD4^+^ T Cells

Naïve T cells were enriched from the spleens of mice using the EasySep Mouse Naïve CD4^+^ T Cell Isolation Kit (Stem Cell Technologies) according to the manufacturer’s protocol. CD11c^+^ DCs were isolated from the BMDC cultures using the IMag Cell Separation System (BD Biosciences) with a biotin anti-mouse CD11c antibody (N418; BioLegend) and BD IMag Streptavidin Particle Plus DM (BD Biosciences). Naïve CD4^+^ T cells (1 × 10^5^ cells/well) were co-cultured with DCs (2 × 10^4^ cells/well) in a 96-well plate. The YB or IL-12 (Peprotech, Cranbury, NJ, USA) was added to the culture medium containing an anti-CD3e antibody (1 μg/mL, 145-2C11; BioLegend) for 3 days, followed by cytokine FACS.

### 4.7. Isolation of Fecal DNA

Feces were non-invasively collected from the mice, immediately frozen in liquid nitrogen, and stored at –80 °C to minimize the change in microbiota composition [51]. DNA was extracted from 1 to 2 feces (≤50 mg) using the ZymoBIOMICS DNA Miniprep kit (Zymo Research, Irvine, CA, USA), according to previous studies [6,7,49,52,53]. The DNA concentration was quantified using the QuantiFluor ONE dsDNA System and Quantus Fluorometer (Promega, Madison, WI, USA).

### 4.8. 16S rRNA Sequencing

16S rRNA sequencing was performed using the MiSeq Reagent Kit according to the guidelines provided by Illumina (San Diego, CA, USA). The 16S rRNA reads were analyzed using Quantitative Insight into Microbial Ecology 2 (QIIME2) Ver.2021.2. The Cutadapt plugin was used to trim the primer region (forward, 17 bases; reverse, 21 bases) from the raw sequences, followed by joining paired-end reads (forward, 250 bp; reverse, 250 bp). Amplicon sequence variants (ASVs) were constructed from the processed reads using the DADA2 algorithm. To perform α- and *β*-diversity analyses, a diversity core-metrics–phylogenetic analysis was used for 10,000 reads set at the sampling depth. Furthermore, the feature classifiers, classify-sklearn, and SILVA138 databases were used to assign the taxonomy. 

### 4.9. Gas Chromatography–Mass Spectrometry (GC/MS) Analysis of Fecal SCFAs

Fecal SCFAs were measured using GC/MS, as previously described [54]. Briefly, the dried feces (per 100 mg) were added to 480 μL of water and 80 μL of 0.5 M phosphoric acid, homogenized using a vortex mixer, and allowed to stand for 24 h in a refrigerator. Subsequently, 60 μL of 0.5 % heptanoic acid as an internal standard, 800 μL of ethanol-methanol (2:8 *v*/*v*), and 80 μL of chloroform–methanol (2:1 *v*/*v*) were added to the mixture, which was homogenized using a vortex mixer. Following this, the suspension was centrifuged at 5000 G for 20 min at 4 °C and the supernatant was passed through a 0.45 μm membrane filter. The resulting solution was analyzed by GC/MS using a Shimadzu GCMS-TQ8030 equipped with a triple quadrupole analyzer and a Supelco Nukol^TM^ column (30 m × 0.25 mm × 0.25 μm). Helium was used as the carrier gas at a flow rate of 0.66 mL/min. The initial oven temperature of 80 °C was maintained for 1 min and then increased to 200 °C at a rate of 15 °C/min. The injected volume was 1 μL and the split ratio was 1:50.

### 4.10. Statistical Analysis

Values are expressed as the mean ± standard deviation (SD). Unpaired *t*-tests (Student’s *t*-tests or Welch’s *t*-tests) were used for comparisons between two groups. R version 3.2.1, Statistical Analysis for Mac version 3.0, and GraphPad Prism 10 were used for the statistical analyses. Differences between more than two groups were analyzed using ANOVA, the post-hoc Dunnett’s multiple comparison test (vs. the control), or Tukey’s HSD test (across groups). Differences were considered statistically significant at *p* values less than 0.05. For multiple comparisons of the 16S rRNA amplicon sequencing data, statistical values (termed *q*-value) were calculated using Welch’s t-test followed by Benjamini and Hochberg’s false discovery rate correction. *q* < 0.05 was used as the threshold for statistical significance.

## Figures and Tables

**Figure 1 molecules-29-01897-f001:**
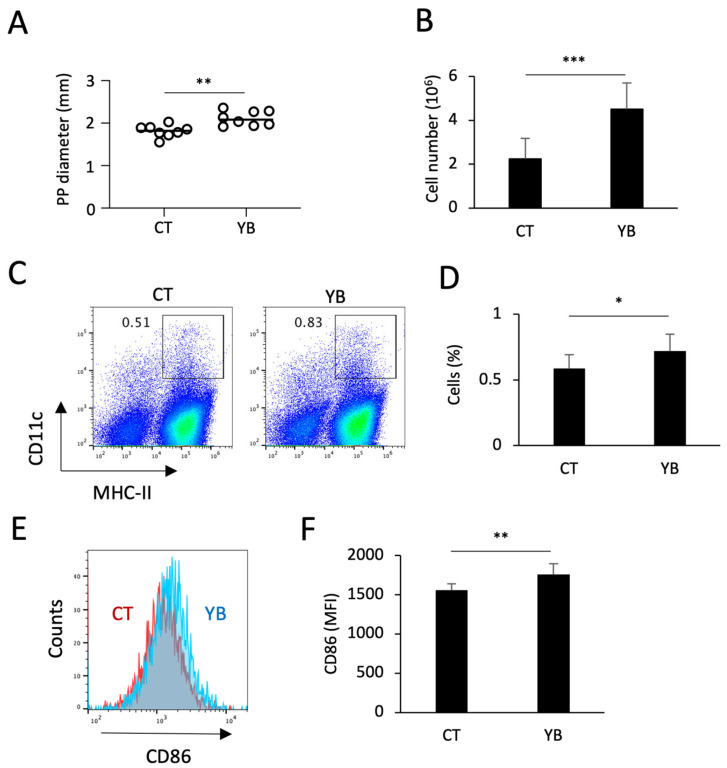
Effect of YB on the cDCs of the PP: (**A**,**B**) the impact of YB on the averaged PP size (diameter) and the number of cells isolated from the PP (n = 8). (**C**,**D**) The percentage of CD11c^+^MHC-II^+^ cDCs in CD45^+^ leukocytes. (**E**,**F**) The expression of CD86 in CD11c^+^MHC-II^+^ cDCs. Values and error bars indicate the mean ± SD. An unpaired *t*-test was used for the statistical analysis. * *p* < 0.05; ** *p* < 0.01; *** *p* < 0.001 versus CT.

**Figure 2 molecules-29-01897-f002:**
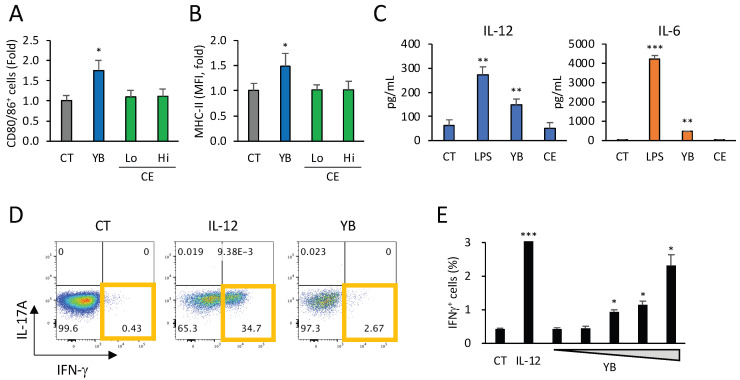
The effects of YB or CE on BMDCs: (**A**) the percentage of CD80/86-positive cells in CD11c^+^ BMDCs treated with YB (3.7 μg/mL) or cellulose (CE, Lo: 1 μg/mL; Hi: 2 μg/mL) (n = 3). (**B**) The expression of MHC-II in CD11c^+^ BMDCs treated with YB or CE (n = 4). (**C**) The concentrations of interleukin (IL)-12, p40, and IL-6 in the culture supernatant of BMDCs in the presence or absence of lipopolysaccharide (LPS: 10 ng/mL, as a positive control), YB (3.7 μg/mL), or CE (2 μg/mL) (n = 3). (**D**,**E**) The effects of YB (0–3.7 μg/mL) on Th1 differentiation in naïve CD4^+^ T cells co-cultured with BMDCs (n = 4). Recombinant IL-12 (10 ng/mL) was used as a positive control for Th1 differentiation. Cytokine concentrations were measured by fluorescence-activated cell sorting using CBA. Values and error bars indicate the means ± SD. Dunnett’s multiple comparison test was used for statistical analysis. * *p* < 0.05; ** *p* < 0.01; *** *p* < 0.001 versus CT.

**Figure 3 molecules-29-01897-f003:**
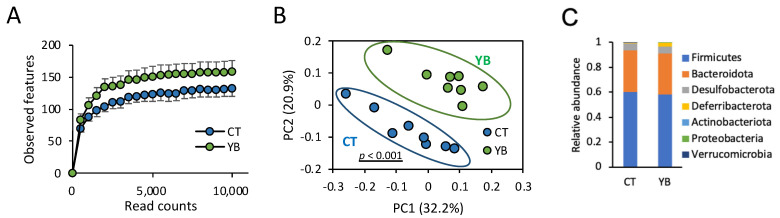
Effect of YB on the gut microbial diversity and composition: (**A**) the impact of YB on the *α*-diversity (observed features) of the gut bacteria analyzed by 16S rRNA sequencing (n = 8). Values and error bars indicate the mean ± SD. (**B**) Principal component analysis of the *β*-diversity values of the gut bacteria (n = 8). (**C**) The effects of YB on the gut bacterial composition at the phylum level (n = 8).

**Figure 4 molecules-29-01897-f004:**
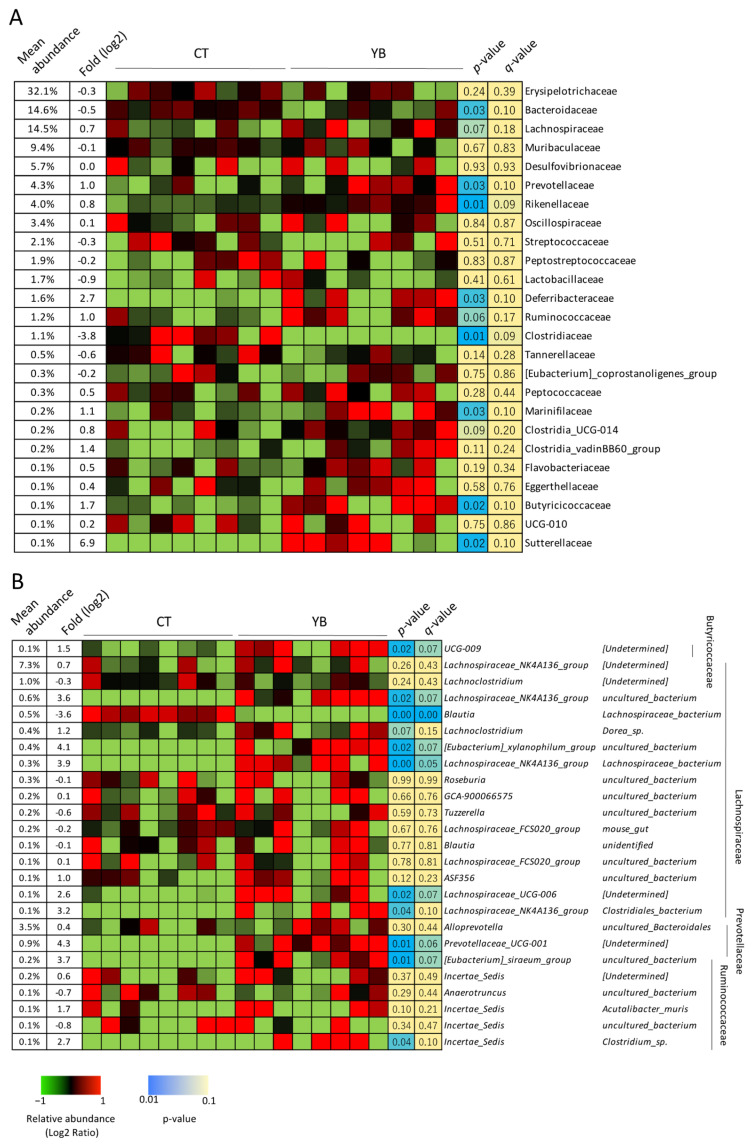
Relative abundance of the fecal bacteria at the family, genus, and species levels: (**A**) the effects of YB on the relative abundance of gut microbiota at the family level (n = 8). Bacteria with a mean relative abundance greater than 0.1% are shown in the heatmap. Blue and light blue represent a statistical significance between the CT and YB groups; yellow indicates no statistical difference. The numbers indicate raw statistical values. (**B**) The effects of YB on the relative abundance of gut microbiota at the genus–species level (n = 8). An unpaired *t*-test was used for statistical analysis.

**Figure 5 molecules-29-01897-f005:**
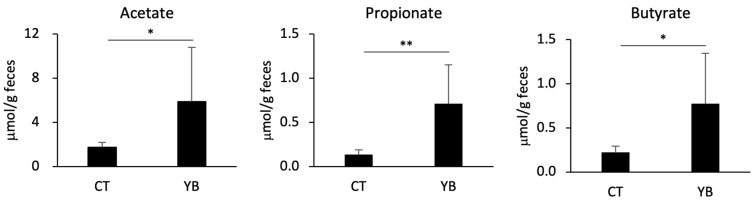
Fecal SCFAs (acetate, propionate, and butyrate) concentration in the YB-fed mice (n = 8). The feces collected from the CT- or YB-fed mice were subjected to GC/MS analysis. Values and error bars indicate the mean ± SD. An unpaired *t*-test was used for the statistical analysis. * *p* < 0.05; ** *p* < 0.01 versus CT.

**Figure 6 molecules-29-01897-f006:**
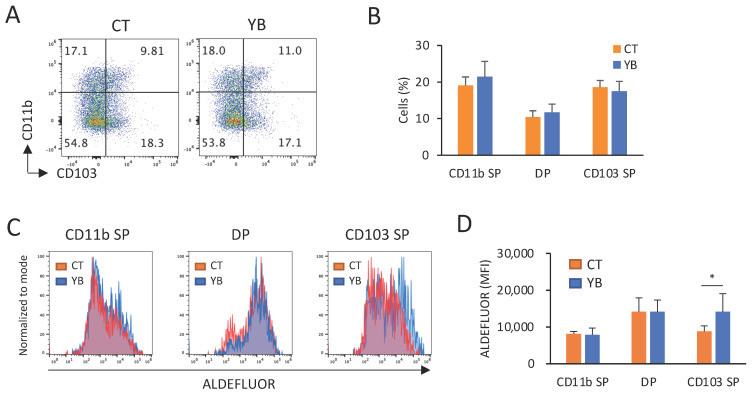
Effect of YB on the cDCs population and RALDH activity in the cDCs of the MLNs. (**A**,**B**) Typical FACS panel and the percentage of CD11b and CD103 expression in MLN CD11c^+^ MHC-II^+^ cDCs (n = 8). (**C**,**D**) Mean fluorescence intensity of ALDEFLUOR, showing RALDH activity, in CD11b SP, CD103 SP, and DP cDCs (n = 8). Values and error bars indicate the mean ± SD. An unpaired *t*-test was used for the statistical analysis. * *p* < 0.05 versus CT.

**Figure 7 molecules-29-01897-f007:**
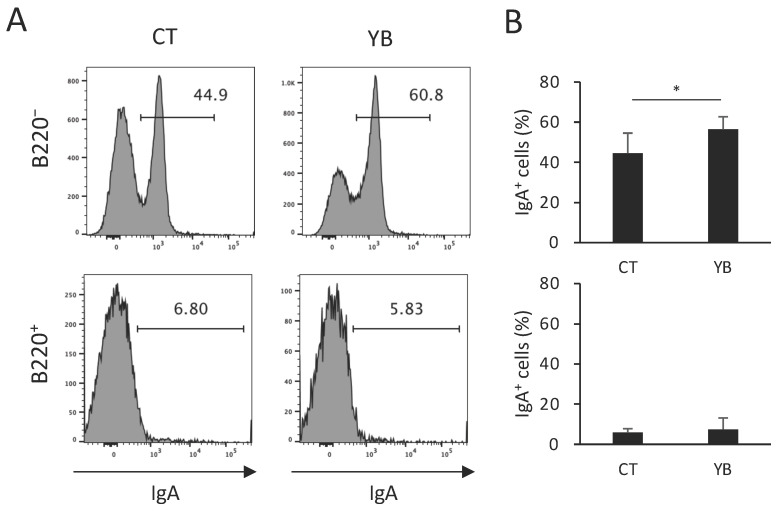
Effect of YB on the IgA^+^ cell population in colonic lamina propria. (**A**) Typical FACS panel showing the percentage of IgA^+^ cells in B220^−^ or B220^+^ cells gated on CD3^−^. (**B**) The percentage of IgA^+^ cells in the B220^−^CD3^−^ or B220^+^CD3^−^ population (n = 8). Values and error bars indicate the mean ± SD. An unpaired *t*-test was used for the statistical analysis. * *p* < 0.05 versus CT.

## Data Availability

Data are contained within the article and Appendix A.

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
