# Peer review of "Beneficial Effects of Dietary Fiber in Young Barley Leaf on Gut Microbiota and Immunity in Mice"

_molecules, 2024, doi:10.3390/molecules29081897_

Round 1
Reviewer 1 Report
Comments and Suggestions for Authors
Overall, the amnuscript submitted by Chudan et al addresses a hot topic by searching for novel health-promoting functional foods/materials via modulating the gut microbiota-intestinal immunity axis, which has emerged as one of the cornerstones of overall physiological homesotasis balancing. The study is well designed and conducted, the idea and the results are clearly presented and in my opinion communicable enough to the wider scientific audiance, not just experts from the field. I would suggest accepting this manuscript after making some of the minor correstions, to be listed below.
Firstly, avoid using the term microbiome throughout the manuscript when speaking about the community of living microorganisms. This ecosystem should always be reffered to as microbiota, since microbiome only reffers to the pool of genetic information contained in this ecosystem. Also, replace 16S rRNA metagenome with 16S rRNA amplicon sequencing - this is a rather techincal issue, but also needs to be corrected.
Next, regarding the results presentation, it could be improved a bit. I am not sure how informative rarefaction curve is in this case. I would prefer to see a boxplot with some of the alpha diversity metrics for a chosen rarefaction depth (10 000 bp). Also, showing taxabarplot at the phylum level is not the best solution as it provides no easily detectable differences. I would prefer to see this barplot at a lower taxonomy level, especially if the authors state that there are significant differences at the level of beta diversity between the two studied groups. What also shuld be improved is filtering out unassigned taxa from the heatmaps presented.
Lastly, the greatest issue for me - but also fixable, is the usage of unpaired ttest for assessing the signifacance of differential abundance of certain bacterial taxa between the samples. Although is sometimes seen that the investigators use this test in microbiota studies with some prior adjustments (e.g. log transform), this does not sufficiently increase the rigour of statistical testing in such cases, simply because it is not apporpriate for compositional data, that arise from microbiota abundance data so it is prone to high false discovery rates. Instead, appropirate tests, that address this issue, like ANCOM, should be used. ANCOM is easy to perfom on your data within the QIIME2 framework, that has already been used as the part of the data analysis pipeline, so I am not sure why did the authors at the first place go around this option? It should definetely be replaced in the revised version of the manuscript. To this end, I must say that, even that this results with loosing statistical significance for some of the taxa, it won`t affect largely the results, as the gut microbiota part with the trends observed in results, definetely fits with the rest of the manuscript.
Author Response
Thank you for your helpful comment and suggestion.
Please see the attached PDF.

Reviewer 2 Report
Comments and Suggestions for Authors
This manuscript describes in detail in an experimental model of mice fed a compound of young barley leaves the effects of this diet on the bacterial structure of the small and large intestine and the effect on the local immune response in both sites with results clearly shown easily. analyzable and understandable, the methodologies used to obtain them are highly reliable and reproducible, although in the discussion I suggest the authors mention their opinion regarding the possible limitations of their design or pending objectives that they could address later in their research. in addition to emphasizing the relevance of their work in the conclusions.
Author Response

(The authors gave the same response as above.)
